# Generation of Flag/DYKDDDDK Epitope Tag Knock-In Mice Using *i*-GONAD Enables Detection of Endogenous CaMKIIα and β Proteins

**DOI:** 10.3390/ijms231911915

**Published:** 2022-10-07

**Authors:** Kazushi Aoto, Shuji Takabayashi, Hiroki Mutoh, Hirotomo Saitsu

**Affiliations:** 1Department of Biochemistry, Hamamatsu University School of Medicine, Hamamatsu 431-3192, Japan; 2Laboratory Animal Facilities & Services, Preeminent Medical Photonics Education & Research Center, Hamamatsu University School of Medicine, Hamamatsu 431-3192, Japan

**Keywords:** flag epitope tag, *i*-GONAD, CRISPR-Cas9 genome editing, AlphaFold2, Western blot, immunohistochemistry

## Abstract

Specific antibodies are necessary for cellular and tissue expression, biochemical, and functional analyses of protein complexes. However, generating a specific antibody is often time-consuming and effort-intensive. The epitope tagging of an endogenous protein at an appropriate position can overcome this problem. Here, we investigated epitope tag position using AlphaFold2 protein structure prediction and developed Flag/DYKDDDDK tag knock-in CaMKIIα and CaMKIIβ mice by combining CRISPR-Cas9 genome editing with electroporation (*i*-GONAD). With *i*-GONAD, it is possible to insert a small fragment of up to 200 bp into the genome of the target gene, enabling efficient and convenient tagging of a small epitope. Experiments with commercially available anti-Flag antibodies could readily detect endogenous CaMKIIα and β proteins by Western blotting, immunoprecipitation, and immunohistochemistry. Our data demonstrated that the generation of Flag/DYKDDDDK tag knock-in mice by *i*-GONAD is a useful and convenient choice, especially if specific antibodies are unavailable.

## 1. Introduction

Genome-editing techniques, such as zinc-finger nuclease (ZFN), transcriptional activator-like effector nuclease (TALEN), and clustered regularly interspaced short palindromic repeats-Cas9 (CRISPR-Cas9), have dramatically reduced the time and cost of generating knockout, knock-in, and conditional knockout mice [1,2,3,4]. Classically, mutant mice are created by gene-targeting embryonic stem (ES) cells and microinjection into fertilized eggs [3,4]. Recently, in vitro electroporation [5,6] and improved genome editing via the oviductal nucleic acid delivery (*i*-GONAD) method [7,8] have been developed in animals, including mice, rats, and hamsters, and show a high productivity of between 50–80% success rate [7,8,9,10]. These methods can insert long DNA fragments, depending on the size of the synthesized single-stranded oligodeoxynucleotides (ssODNs), by homology-directed repair (HDR) mechanisms of double-strand breaks. Ohtsuka et al. [8] synthesized 1 kb long ssODNs from a messenger RNA template and developed knock-in mice with a fluorescent protein, mCitrine, but with a lower success rate of 15%. For most researchers, it is convenient to use commercially available 130 and 200 base pairs (bp) ssODNs. Since the length of homology arms for HDR needs to be more than the total length of 75–85 bp [11,12], it is possible to design homology arms within 70 bp or 140 bp for *i*-GONAD of the epitope tag.

Various commercially available and popular epitope tags have been used for knock-in mouse analysis, including the Flag/DYKDDDDK tag (8 amino acids (AAs)) of synthetic epitope tag [13,14,15], HA tag (YPYDVPDYA, 9 AA) of AA residues 98–106 of human influenza hemagglutinin [16,17], Myc tag (EQKLISEEDL, 10 AA) of C-terminal (AA residues 410–419) of human c-myc protein [18], V5 tag (GKPIPNPLLGLDST, 14 AA or shorter version IPNPLLGLD, 9 AA) of AA residues 95–108 of RNA polymerase alpha subunit of simian virus 5 [4], and fluorescent tags (GFP, RFP, etc.) [4]. Flag tags are commonly used for immunoprecipitation (IP), Western blotting (WB), and immunohistochemistry (IHC) in knock-in mice [13,14,15]. In many cases, epitope tag knock-in mice have been created by pronuclear microinjection into fertilized eggs [4,13,14,15,16]. The position of an epitope tag is important for protein function.

Calcium/calmodulin-dependent protein kinase type II subunit alpha and beta (CaMKIIα and β) regulate neuronal plasticity and memory [19], and are related to human disease and autosomal dominant intellectual disorders [20,21]. The three-dimensional (3D) structure and functional domain of CaMKIIα/β have already been reported [22]. It has been reported that in vivo Flag and HA tagging into the N or C-terminus of CaMKIIα and β by in utero electroporation enables the rapid determination of the localization of endogenous CaMKIIα and β proteins in the mouse brain [23]. However, epitope tag knock-in mice with CaMKIIα/β have not yet been produced.

In this study, we investigated the knock-in position of the Flag/DYKDDDDK tag based on the 3D model of protein structure predicted by the AlphaFold2 program [24,25] and developed CaMKIIα and CaMKIIβ knock-in mice using *i*-GONAD. Flag-CaMKIIα and-β proteins were confirmed using biological and histological assays. Our data showed that the Flag/DYKDDDDK tag knock-in mouse technique had advantages over a specific antibody-based assay.

## 2. Results

### 2.1. Investigation of Flag/DYKDDDDK Epitope Tag Position by AlphaFold2 Protein Structure Database

In the generation of epitope knock-in mice, we first determined the epitope tag position for the target protein. The epitope tag is a small peptide with a size of 8 to 14 AA, which can disrupt the target protein function when placed near the functional domain or localization domain in some cases [26,27]. CaMKIIα and β proteins have a kinase domain in the N-terminal, a regulatory segment, a linker region in the middle, and an association domain in the C-terminal [22] (Figure 1a). We investigated the 3D structure of Flag-CaMKIIα and Flag-CaMKIIβ modeled by ColabFold: AlphaFold2 using MMseqs2 software (https://colab.research.google.com/github/sokrypton/ColabFold/blob/main/AlphaFold2.ipynb) (accessed on 16 March 2022) [24,25]. N-terminal peptides are well situated on the protein surface, and Flag peptides are very likely to be accessed by the anti-DYKDDDDK antibody (Figure 1b,c). Thus, the Flag/DYKDDDDK epitope tag at the N-terminal of CaMKIIα and β appears to be promising for the generation of the epitope tag knock-in mice.

### 2.2. Flag Knock-In Mouse Generation Using i-GONAD Method

To generate Flag/DYKDDDDK epitope tag knock-in mice, guide RNAs (gRNA) for CaMKIIα and β were selected using CRISPR Targets track of the UCSC Genome Browser (https://genome.ucsc.edu) (accessed on 9 December 2017). Epitope tag knock-in mice were generated using *i*-GONAD. The HiFiCas9 protein, CRISPR RNA (crRNA), trans-activating crRNA (tracrRNA), and ssODNs were mixed, and the mixed solution was injected into the oviduct ampulla using a mouth glass capillary. After electroporation with an NEPA21 electroporator and electrode, the oviduct ampulla, ovary, and uterus were returned to the intra-abdominal cavity.

For *CaMKIIα^N-Flag^* knock-in mice, six pups were obtained from four mothers: two wild-type pups, three knock-in pups, and one with deletion of several bases. For *CaMKII**β^N-Flag^* knock-in mice, a total of thirteen pups were born from four mothers: four wild-type pups, six knock-in pups, and three pups with deletions (Figure 2g). All pups were genotyped by Sanger sequencing using specific primer sets around the knock-in region (Figure 2b,c,e,f and Appendix A). *CaMKIIα^N-Flag^* and *CaMKII**β^N-Flag^* knock-in mice were crossed with outbred ICR mice for more than three generations to avoid potential off-target changes caused by CRISPR-Cas9 genome editing. Heterozygous (*N-Flag/+*) and homozygous (*N-Flag/N-Flag*) mice of *CaMKIIα^N-Flag^* and *CaMKII**β^N-Flag^* did not show different body sizes and lengths, compared with wild-type mice. Moreover, both homozygous mice could mate with wild-type mice, indicating that the reproductive functions of the Flag-CaMKIIα and Flag-CaMKIIβ mice were normal. Thus, if a suitable tag position is selected using AlphaFold2, tag knock-in mice would reduce or minimize the dysfunction of the target gene.

### 2.3. Biochemical Assay Using CaMKIIα^Fla^ and CaMKIIβ^Flag^ Mice

To assess whether N-terminal Flag-tagged CaMKIIα and β proteins were produced, we performed WB using commercially available anti-Flag/DYKDDDDK monoclonal antibody (clone 2H8) [28] in cortical, hippocampal, and cerebellar tissues of wild-type, *CaMKIIα^N-Flag^* and *CaMKIIβ^N-Flag^* mice. Flag-CaMKIIα protein was strongly expressed in the cortex and hippocampus but weakly expressed in the cerebellum (Figure 3a). Flag-CaMKIIβ was strongly expressed in the cortex, hippocampus, and cerebellum (Figure 3a). These results are similar to those of previous studies [29]. As expected, wild-type ICR mice did not show any specific bands. Next, as it is well known that CaMKII forms a dodecamer, in which α and β subunits are predominant in the brain [22], we performed IP and WB using a cortical lysate of *CaMKIIα^N-Flag/N-Flag^* homozygous knock-in mice to examine whether Flag-CaMKIIα is associated with endogenous CaMKIIβ proteins. Flag-CaMKIIα bound to the endogenous CaMKIIβ protein (Figure 3b). The Flag-CaMKIIα protein showed a slightly shifted band, compared to the wild-type band, and its expression level was comparable to that of endogenous CaMKIIα (Figure 3b, asterisk). The expression levels of CaMKIIβ were also comparable between wild-type (WT) and *CaMKIIα^N-Flag/N-Flag^* mice. These results indicate that Flag knock-in mice show a similar expression profile to that of endogenous CaMKIIα and β, and Flag-CaMKIIα works as a heterogeneous complex of probable dodecamers.

### 2.4. Immunostaining Using CaMKIIα^N-Flag^ and CaMKIIβ^N-Flag^ Mice

Next, to examine the localization of Flag-CaMKIIα and β proteins in the brain, immunostaining using anti-DYKDDDDK and anti-neuronal nuclei (NeuN) antibodies was performed on hippocampal CA1, cortex, and cerebellum of wild-type, *CaMKIIα^N-Flag/+^* heterozygous, and *CaMKIIβ^N-Flag/+^* heterozygous knock-in mice. Flag-positive cells were not found in wild-type ICR mice (Figure 4a,d,g). Flag-CaMKIIα protein was strongly expressed in the neuronal cell bodies and axon/dendrites of the hippocampal CA1 and cortical layers, especially in the cell body and dendrites of cerebellar Purkinje cells, but not in the molecular layer and granular layer (Figure 4b,e,h). Flag-CaMKIIβ protein was expressed in hippocampal and cortical neural cells and in all cell types of the cerebellum (Figure 4c,f,i). These results are similar to the published expression patterns of CaMKIIα and β [30]. Moreover, Flag/DYKDDDDK staining in the hippocampal CA1 and dentate gyrus completely overlapped with the anti-CaMKIIα antibody staining in *CaMKIIα^N-Flag/+^* mice (Figure 4j–o). In this case, the anti-CaMKIIα antibody detected endogenous CaMKIIα and Flag-CaMKIIα proteins. These results indicate that Flag knock-in for CaMKIIα and CaMKIIβ enables the detection of FLAG-tagged proteins using commercially available epitope-specific antibodies.

## 3. Discussion

Epitope tag knock-in mice are useful for protein expression and localization analysis. In expression vector or knock-in experiments, tag positions of target proteins are mainly in the N- or C-terminus of the target protein and in the middle, except for the functional domain and signal domain in very few cases [17,27]. This selection should be based on information on the 3D structure of the target protein. Our prediction method for determining the position of epitope tag knock-in using AlphaFold2 takes advantage of AlphaFold2-enabled highly accurate prediction of protein structures [24]. Our prediction results (Figure 1) were similar to the published 3D structural information for CaMKIIα and β and its dodecamer [22]. The N- and C-terminal portions are shown in red in the 3D structure with a low confidence score (pLDDT), but both portions that are not alpha-helical bundles and beta-sheet domains should be considered as flexible protein structures that are amenable to tagging. AlphaFold2 also contributes to several biological processes, such as protein–protein complexes [31], prediction of the transcriptional activator domain [32], pathogenicity of missense variants [33,34], and phenotypic effects of single mutations [35]. Thus, our prediction for flag/DYKDDDDK epitope tag knock-in mouse is an application that uses structural modeling with AlphaFold2 and is able to escape the disruption of protein function by epitope tag.

The *i*-GONAD method is a powerful tool for rapidly producing CRISPR-Cas9 genome-edited mice and successfully producing N-terminal Flag/DYDDDDK tag knock-in mice against CaMKIIα and β (Figure 2). In many epitope tag knock-in mice, the number of the epitope tag is increased as a tandem repeat of twice or triple for increasing epitope recognition by antibody. In this study, we used a single Flag/DYDDDDK tag because the anti-DYKDDDDK monoclonal antibody, 2H8 clone, has high affinity for Flag/DYDDDDK tag protein sequence, but it can only be used for the detection of N-terminally tagged proteins [28]. One small tag may be preferable in terms of the potential risk of affecting target protein function by tagging and the higher success rate of knock-in because the success rate of gene knock-in would decrease depending on the increasing fragment size [36,37] and on the decreasing homology arm length [36,38].

The *i*-GONAD is a powerful method with high efficiency for knockout [6,8], or knock-in of short epitope tags and pathological single nucleotide variants (SNV) [6,8] and conditional knockout with loxP sequence [39] in mice (Figure 5, green round). The success rate of EGFP knock-in mice using synthesized ssODNs from a messenger RNA template was low at 15% [8] (Figure 5, orange round). However, to generate long insertional knock-in mice using more than 1 kb for many fluorescent proteins, reporter genes, Cre/Flippase recombinase, and replacement of human genes, we needed to develop a large fragment knock-in system (Figure 5, yellow round). Recently, an adeno-associated virus 1 or 6 (AAV1, AAV6)-mediated single-strand DNA delivery system for providing knock-in donors was shown to pass the zona pellucida surrounding the mouse oocyte and infect fertilized eggs [40,41]. Alternatively, the creation of a sufficient amount of long single-stranded strand DNAs would be useful, and double-strand DNA delivery by a gene targeting vector using the *i*-GONAD method would be much better if possible. Such technical developments would probably be able to create long-fragment knock-in mice using the *i*-GONAD method.

## 4. Materials and Methods

### 4.1. AlphaFold2 Analysis for Prediction of Flag Epitope Tag Position

The suitability of the FLAG epitope tag position for CaMKIIα and CaMKIIβ was estimated from the 3D structure of the proteins. The prediction of both protein structures was performed using ColabFold: AlphaFold2 with MMseqs2 (https://colab.research.google.com/github/sokrypton/ColabFold/blob/main/AlphaFold2.ipynb) (accessed on 16 March 2022) [24,25], CaMKIIα (NP_803126.1), and CaMKIIβ protein sequence (NP_001167524.1). The N-terminal Flag peptide sequence, DYKDDDDK, was added just after methionine protein of the translational initiation protein, similar to that of Flag/DYDDDDK tag knock-in mice.

### 4.2. Flag Knock-In Mouse Generation Using i-GONAD Method

The *i*-GONAD method was performed, as previously described [7,8]. Experiments were performed in accordance with the guidelines of the Hamamatsu University School of Medicine Committee on Recombinant DNA Security (No. 3-25). All experiments were approved by the Hamamatsu University School of Medicine Animal Care and Use Committee (No. 2020019). ICR mice were purchased from Japan SLC, Inc. (Shizuoka, Japan). ICR mice were mated at evening 4–6 pm, and the next day, they were checked for plugs. In the evening time (4–6 pm) on the same day, plugged females were injected with CRISPR-Cas9 solution into the oviduct ampulla that was operated from the abdomen. The gRNA targets were selected as green in the UCSC genome browser (https://genome.ucsc.edu) (accessed on 9 December 2017). All CRISPR materials were purchased from Integrated DNA Technology, Inc. (IDT, Coralville, IA, USA) and Macrogen Inc. (Seoul, Korea) as the following name and sequence of gRNA and ssODN. CaMK2A-Ex1-gRNA: CTGCCTGCCCAGTGCCAGGA. CaMK2B-Ex1-gRNA: GAGCCCGATCGCCACCGCCA. Flag-CaMKIIα-ssODN: CCCAGCCCTAGTTCCCAGCCTAAAGCCTCGCCTGCCTGCCCAGTGCCAGGATGGATTACAAGGATGACGATGACAAGGCTACCATCACCTGCACCCGATTCACAGAAGAGTACCAGCTCTTTGAGGAACT. Flag-CaMKIIβ-ssODN: GCCGCCGCCCGAGCGCAGCCGAGCGCACGCCGAGCCCGATCGCCACCGCCATGGATTACAAGGATGACGATGACAAGGCCACCACGGTGACCTGCACCCGTTTCACCGACGAGTACCAGCTATACGAGGA.

CRISPR-Cas9 solution contained as 30 μM crRNA, 30 μM tracrRNA, 1 μ/μL HiFiCas9 protein, 1 μ/μL ssODN, and 0.02% Fast Green in Opti-MEM (ThermoFisher, 31985062, Waltham, USA) and was filled into a glass capillary and injected into the oviduct ampulla in a total 1μL volume. After injection, the oviduct ampulla was covered with phosphate-buffered saline (PBS)-soaked KimWipes (Kimberly Clark Corp, Irving, USA) and was electroporated using an electrode under the following conditions (Porlin pulse: Voltage 50 V, pulse length 5 ms, pulse interval 50 ms, No. pulses 3, decay 10%, polarity +/−, transfer pulse: Voltage 10 V, pulse length 50 ms, pulse interval 50 ms, and no. pulses 3, Decray 40%, Polarity +/−) using a NEPA21 electroporator and tweezers without variable gap electrodes for Oviduct, CUY652P2.5X4 (NEPA GENE Co., Ltd. Ichikawa city, Japan).

### 4.3. Genotyping of Flag-CaMKIIα and Flag-CaMKIIβ

Genotyping PCR primers around the translational starting site of exon1 in the *CaMKIIα* and *CaMKIIβ* genes were selected based on the following names and sequences. PCR conditions was performed as 94 °C for 2 min, 30 cycles (98 °C for 10 s, 55 °C for 30 s, 68 °C for 30 s), 68 °C for 5 min. The wild-type band was 105 bp. The flag knock-in band was 129 bp using the following primer set. CaMK2A-Ex1-105F: TCAGCATCCCAGCCCTAGTTCCCAG. CaMK2A-Ex1-105R: CATGCTGCACACCTCCCTCTGCATG. CaMK2B-Ex1-100F: AGCCCGATCGCCACCGCCAT. CaMK2B-Ex1-100R: CGCGCCGAGGCTCTTACTTGC. CaMK2A-454F: CAGGCAGGTGTTGGGGAGGCAGTTA. CaMK2A-454R: CATGCTGCACACCTCCCTCTGCATG. CaMK2B-376F: GCGGGGACAGAGCGAGCAGATCTC. CaMK2B-376R: CTTGGGCCCGCAGCTGGAAGGAG.

### 4.4. Immunoblotting and Immunoprecipitaion

Brain tissues of *CaMKIIα^Flag/+^* and *CaMKIIβ^Flag/+^* mice at 8–12 weeks were collected, snap frozen in liquid nitrogen, and then stored at −80 °C until finishing genotyping and starting WB. Samples were lysed with 500 μL RIPA buffer (150 mM NaCl, 50 mM Tris-HCl pH 7.4, 1 mM EDTA pH 8.0, 1% NP-40, 0.5% sodium deoxycholate, 0.1% SDS) supplemented with cOmplete Mini, EDTA-free Protease Inhibitor Cocktail Tablets (Roche, 11836170001, Basel, Switzerland through 10 passages with a 21G needle and syringe. After rotating on shaker for 30 min at 4 °C and centrifugation at 9100× *g* for 10 min at 4 °C, supernatant samples were mixed with 2× SDS sample buffer (0.125 M Tris-HCl pH 6.8, 4% SDS, 20% glycerol, 0.01% BPB, 100 μL/mL 2-mercaptoethanol). For immunoprecipitation, the lysis solution was incubated with mouse monoclonal anti-DYKDDDK (clone 2H8) (1/200 dilution, TransGenic Inc., Fukuoka, Japan), and Dynabeads ProteinG (VERITAS, DB10004, Tokyo, Japan). Boiled samples were separated by SDS-PAGE and transferred to a PVDF membrane (Clear Trans PVDF membrane, Hydrophobic, 0.45 μm, 034-25663, FUJIFILM) with EzFastBlot (2332590, ATTO, Tokyo, Japan) using a semi-dry blotter (WSE-4115 PoweredBlot Ace, ATTO Corporation, Japan). After the transfer, the membranes were fixed with 4% paraformaldehyde (PFA, P6148-500G, Sigma-Aldrich, St. Louis, USA)/PBS for 30 min to increase the band signal. The membranes were blocked with 3% milk (Skim Milk Powder, 198-10605, FUJIFILM Wako Pure Chemical Corp., Osaka, Japan)/TBST (TBS + 0.1% Tween-20), and then incubated with the following primary antibodies: mouse monoclonal anti-CaMKIIα (1/1000 dilution, 6G9 clone, #50049, Cell Signaling, Danvers, USA), rabbit anti-CaMKIIβ (1/2000 dilution, GTX133072, Gene Tex, Irvine, CA, USA), mouse monoclonal anti-DYKDDDK (clone 2H8, 1/1000 dilution, TransGenic Inc., Japan), and mouse anti-GAPDH (1/50,000 dilution, 60004-1-Ig, Proteintech, Rosemont, USA) antibodies at 4 °C for overnight. The next day, after washing three times with TBST for 10 min, the PVDF membrane was incubated with the following secondary antibodies: goat anti-mouse and anti-rabbit antibodies conjugated with horseradish peroxidase (1/5000 dilution, Jackson ImmunoResearch, West Grove, PA, USA) for 1 h. After washing three times with TBST for 10 min, blotted membranes were detected using the Clarity Western ECL Substrate (#170-5060, Bio-Rad Laboratories. Inc., Hercules, FL, USA) for 5 min and FUSION-FX7. The EDGE Chemiluminescence Imaging System (M&S Instruments, Inc., Osaka, Japan).

### 4.5. Immunohistochemistry

We analyzed two or more independent mice for each genotype. After transcardial perfusion with PBS and 4% PFA/PBS, the brains of FLAG tag knock-in mice at 8–12 weeks were dissected and fixed in 4% PFA/PBS solution overnight at 4 °C. After washing with cold PBS, the samples were dehydrated using an ethanol/xylene series and embedded in paraffin. Sectional samples were cut at 5 μm thickness using a microtome (ThermoFisher HM430, USA) and placed on glass slides (MTSUNAMI, Platinum Pro, PRO-04, Osaka, Japan). After deparaffinization with xylene and 100% ethanol, and washing with 70% ethanol and running water for each 5 min, antigen retrieval was performed on sections with 10 mM sodium citrate solution with microvan for 10 min. A Super PAP pen Liquid Blocker (Daido Sangyo) was used to draw a hydrophobic circle around the brain tissue on a slide. After blocking with 3% BSA (albumin, from bovine serum, protease free, 018-15154, WAKO, Japan)/PBT (PBS plus 0.1% Tween-20 (sc-29113, Santa Cruz Biotechnology, Dallas, TX, USA) on the glass slides for 1 h at RT, sectional slides were incubated overnight at 4 °C with either of the following primary antibodies: mouse monoclonal anti-DYKDDDK (clone 2H8, 1/200, TransGenic Inc., Fukuoka, Japan) or rabbit anti-NeuN (1/500, ab177487; Abcam, Cambridge, UK). After washing three times with PBS, tissue slides were incubated for 1 h at RT with Alexa Fluor 488 or 546 donkey anti-mouse IgG and anti-rabbit IgG (1/200–1/400 dilution, A10036, Thermo Fisher Scientific, Waltham, MA, USA) as the secondary antibodies. After washing three times with PBS, the tissues were stained with the nuclear marker DAPI (1/1000 dilution, DOJINDO, NX034, Masushiro, Japan) and mounted with aqueous mounting media (Diagnostic BioSystems, Fluoromount/Plus, K048, Pleasanton, CA, USA). The mounted slides were imaged using a 20× objective lens on a Leica TSC SP8 confocal microscope (Leica Microsystems, Wetzlar, Germany).

## Figures and Tables

**Figure 1 ijms-23-11915-f001:**
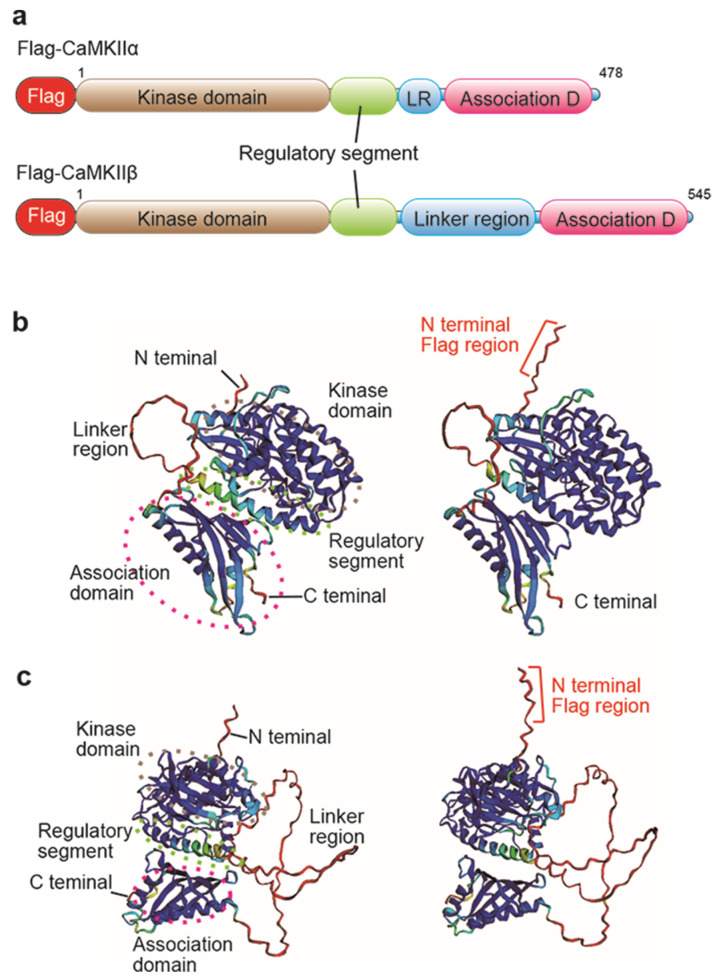
Flag-CaMKIIα and-β protein structures predicted by AlphaFold2. (**a**) Flag-tagged CaMKIIα and CaMKIIβ protein structures with a Flag tag (red), kinase domain (brown), regulatory segment (green), linker region (LR, blue), and association domain (pink). (**b**) 3D structure of CaMKIIα (left) and Flag-CaMKIIα (right) proteins by ColabFold. (**c**). The 3D structure of CaMKIIβ (left) and Flag-CaMKIIβ (right) proteins using ColabFold. The prediction quality as per residue confidence score, pLDDIT, is shown as colors of 3D structure, very high (blue, pLDDT > 90), confidence (light blue, 90 > Plddt > 70), low (yellow, 70 > pLDDT > 50), and very low (red, pLDDT > 50).

**Figure 2 ijms-23-11915-f002:**
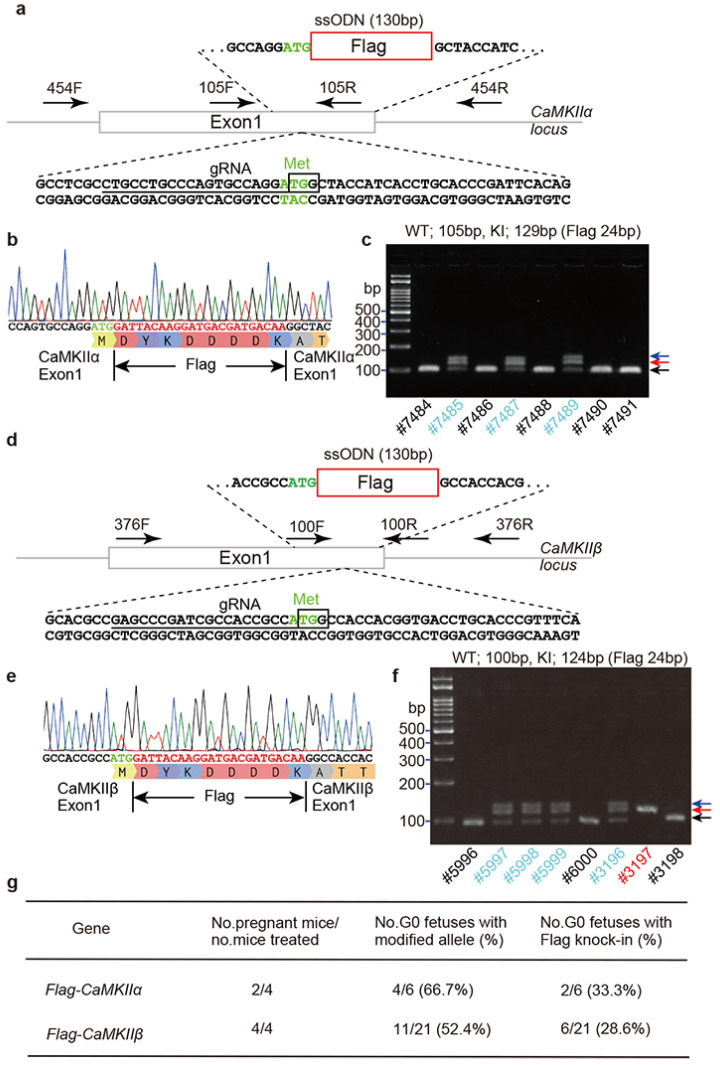
Generation of Flag knock-in mice for *CaMKII**α* and *CaMKII**β*. (**a**) Sequence of exon 1 of mouse *CaMKII**α*, gRNA target site (black underline), and ssODN. The square box shows protospacer adjacent motif (PAM) sequence. The start codon encoding methionin (Met) is highlighted with green. (**b**) Confirmation of knock-in sequences. PCR products using genomic DNA as a template were cloned and sequenced. (**c**) Gel electrophoresis of PCR products amplified with primer sets of CaMKIIα-Ex1-105F/R in (**a**). Black and blue colored IDs indicate wild-type (WT) and knock-in (KI) pups, respectively. Black, red, and blue arrows show wild-type (105 bp), KI (129 bp) and heteroduplexed band of wild-type and KI DNA fragments, respectively. (**d**–**f**) Sequence of exon 1 of mouse *CaMKII**β*, gRNA target site (black underline), and ssODN. Square box shows PAM sequence. The start codon encoding methionin (Met) is highlighted with green. (**e**) Confirmation of knock-in sequences. (**f**) Gel electrophoresis of PCR products amplified with primer sets of CaMKIIβ-Ex1-100F/R in (**d**). Black- and blue-colored IDs indicate wild-type (WT) and knock-in (KI) pups, respectively. Black, red, and blue arrows show WT (100 bp), KI (124 bp) and heteroduplexed band, respectively. (**g**) A summary table of generating flag knock-in mice by *i*-GONAD.

**Figure 3 ijms-23-11915-f003:**
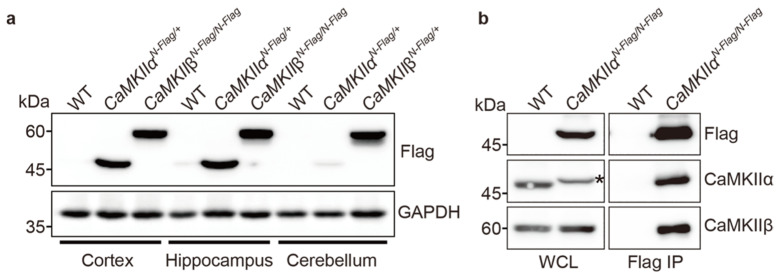
Expression of Flag-CaMKIIα and β and association between Flag-CaMKIIα and endogenous CaMKIIβ. (**a**) Immunoblotting of cortical, hippocampal, and cerebellar samples in *CaMKII**α^N-Flag/+^* heterozygous and *CaMKIIβ^N-Flag/N-Flag^* homozygous knock-in mice using anti-Flag antibody. GAPDH was used as internal control. (**b**) Immunoblotting using anti-Flag, CaMKIIα, and CaMKIIβ specific antibodies, after immunoprecipitation of cortex sample in *CaMKII**α* mouse using flag antibody (Flag IP). WCL, whole cell lysate. Asterisk shows shifted band of Flag-CaMKIIα protein, compared to that of endogenous CaMKIIα protein.

**Figure 4 ijms-23-11915-f004:**
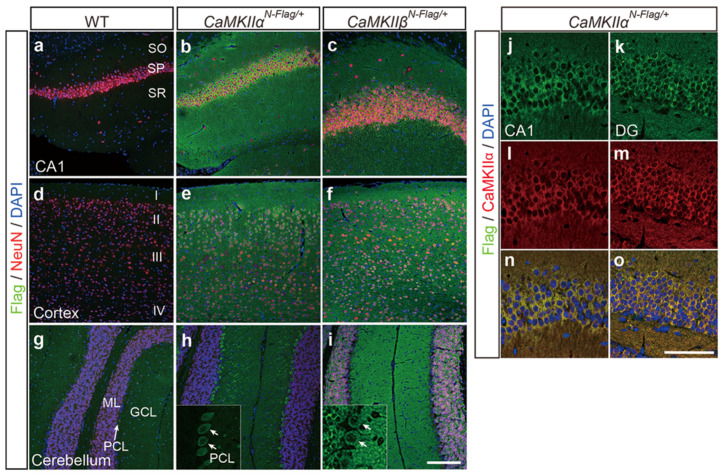
Detection of Flag-CaMKIIα and Flag-CaMKIIβ by immunohistochemistry. Immunostaining of hippocampal CA1 region (**a**–**c**), cortex (**d**–**f**), and cerebellum (**g**–**i**) of wild-type (WT, (**a**,**d**,**g**)), *CaMKII**α^N-^**^Flag^* (**b**,**e**,**h**), and *CaMKII**β^N-Flag^* (**c**,**f**,**i**) using anti-Flag (green) and neural marker anti-NeuN (red) antibodies. DAPI (blue) is nuclear marker. SO, stratum oriens, SP, stratum pyramidal, SR, stratum radiatum, SLM, stratum lacunosum-moleculare. Cortical layer I, II, III, and IV in D. ML, molecular layer, PCL, Purkinje cell layer, and GCL, granule cell layer. (**j**–**o**) Staining of anti-CaMKIIα and anti-Flag/DYKDDDDK antibodies showed complete overlap in cytoplasmic region of hippocampal CA1 (**j**,**l**,**n**) and dentate gyrus (DG) (**k**,**m**,**o**) of *CaMKII**α^N-Flag/+^* mice. Scale bar = 100 μm in (**a**–**i**), 50 μm in (**j**–**o**).

**Figure 5 ijms-23-11915-f005:**
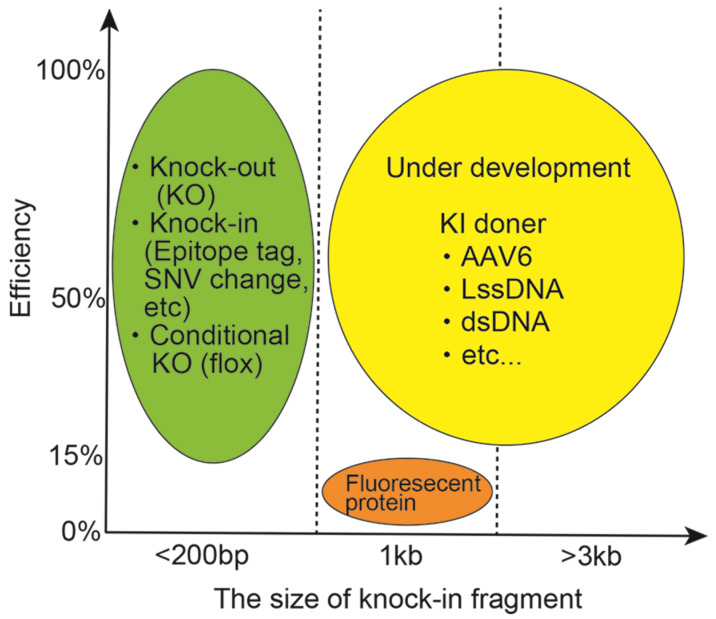
**The***i*-GONAD efficiency for knock-out (KO), knock-in, and conditional KO. Sematic representation of genome editing efficiency and the size of knock-in fragment.

## Data Availability

All data are available within the article and Supplementary Information. The materials of this study are available from the corresponding author upon request.

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
