# Peer review of "Generation of Flag/DYKDDDDK Epitope Tag Knock-In Mice Using i-GONAD Enables Detection of Endogenous CaMKIIα and β Proteins"

_ijms, 2022, doi:10.3390/ijms231911915_

Round 1
Reviewer 1 Report
This manuscript reported the generation of both CaMKIIα and CaMKIIβ epitope tagging mice using CRISPR-assisted ssODN-mediated HDR. The authors successfully obtained knock-in mice with FLAG tagged on the N-terminus of both CaMKIIα and CaMKIIβ. These mice enabled localization and expression analysis using FLAG antibody. I have the following concerns:
1. Based on figure 3 and 4. it seems the localization and expression can also be achieved by CaMKII-specific antibodies. The authors need to demonstrate further the advantage of epitope tagging. Is there any potential applications of epitope tagging that heavily rely on the use of epitope-specific antibodies?
2. The authors failed to provide sufficient information on the gene editing assay. All primers should be clearly indicated in figure 2a regarding their relative location on the genome as well as the ssODN template. The detailed genotypes of the knock-in mice also need to be added. In addition, the 5’ and 3’ junction should be sequenced to demonstrate the accuracy of the gene editing.
3. In figure 2b, PCR products were cloned and only expected gene edited amplicon was showed. The authors may want to provide a deep sequencing result of the PCR product to show a whole profile of the gene editing events because it has been observed that incomplete HDR could occur when ssODN was used as HDR template in porcine gene editing studies.
4. Can you show some evidence demonstrating the identity of the band indicated by the red arrow in Figure 2. How can you confirm it represents a heteroduplexed band of wild-type and KI DNA fragments.
5. Line 103-104. Overstate. First, the N-terminal epitope tagging is different from gene disruption. Second, a systematic enzymatic evaluation of the effect of epitope tagging need to be performed other than analyzing localization and body size if the authors want to make the claim.
6. Line 44 and 67. Rephrase please.
Author Response
Reviewer #1 (Remarks to the Author):
We would like to thank the reviewer #1 for your instructive comments. We revised our manuscript according to your comments. We made corrections with "Track changes while editing" features of MS Word so that you can easily see which changes have been made. We worked with an editing company to improve our manuscript.
- Based on figure 3 and 4. it seems the localization and expression can also be achieved by CaMKII-specific antibod The authors need to demonstrate further the advantage of epitope tagging. Is there any potential applications of epitope tagging that heavily rely on the use of epitope-specific antibodies?
Thank you for your important comments. As shown in Figure 3b, Flag tagging enable us to efficiently precipitate Flag-CaMKIIα by using anti-Flag antibody. This finding suggested the usefulness of tag knock-in mice for association assay between tagged protein and DNA, RNA, and protein. Proteomics assay following immunoprecipitation is one of the potential applications because antibodies that can work well for immunoprecipitation are limited. Concerning specific application for CaMKIIαN-Flag and CaMKIIβN-Flag mice, phosphoproteomics using immunoprecipitated samples may be applicable to identify target proteins that would be phosphorylated by CaMKIIα/β with ionophore treatment.
- The authors failed to provide sufficient information on the gene editing assaAll primers should be clearly indicated in figure 2a regarding their relative location on the genome as well as the ssODN template. The detailed genotypes of the knock-in mice also need to be added. In addition, the 5’ and 3’ junction should be sequenced to demonstrate the accuracy of the gene editing.
We made new figures 2a and 2d. We also checked the 5’ and 3’ junction regions of ssODNs by PCR using a primer set of 454F/R in CaMKIIα and 376F/R in CaMKIIβ (Figure 2). We also added the sequence of these primers in the Materials and Methods (Line 296-299). We found that homology directed-repair was precisely occurred in both CaMKIIαN-Flag and CaMKIIβN-Flag mice (Supplementary Figure 1). However, we found that a 1-bp insertion in 5’-UTR of CaMKIIβ (Supplementary Figure 1), suggesting synthesis error of ssODN. We regret that we did not check the sequences. Fortunately, because the insertion is likely not to affect any upstream open reading frame in the 5’-UTR, the effect of the deletion on translation would be negligible.
- In figure 2b, PCR products were cloned and only expected gene edited amplicon was showed. The authors may want to provide a deep sequencing result of the PCR product to show a whole profile of the gene editing events because it has been observed that incomplete HDR could occur when ssODN was used as HDR template in porcine gene editing studies.
Thank you for your important suggestion. However, because DNA of the founder mouse is unavailable, we did not perform a deep sequencing of the PCR product.
- Can you show some evidence demonstrating the identity of the band indicated by the red arrow in Figure 2. How can you confirm it represents a heteroduplexed band of wild-type and KI DNA fragments.
As shown in the figure below, there are three bands indicated by black (wild-type allele), red (Flag knock-in allele), and blue arrows only in the heterozygous mouse. T7 Endonuclease I (T7E1) digestion, which selectively cut a heteroduplexed band, showed that the uppermost band indicated by the blue arrow was digested, indicating that the band represented a heteroduplexed fragment.
- Line 103-104. Overstate. First, the N-terminal epitope tagging is different from gene disruption. Second, a systematic enzymatic evaluation of the effect of epitope tagging need to be performed other than analyzing localization and body size if the authors want to make the claim.
We completely agree with your comment and changed to “tag knock-in mice would reduce or minimize the dysfunction of the target gene” on line 112-113.
- Line 44 and 67. Rephrase please.
Line 44. We changed to “it is possible to design homology arms within 70 bp or 140 bp for the epitope tagging using i-GONAD” on line 45-47.
Line 67. We changed to “Flag/DYKDDDDK tag knock-in mouse technique had advantages over specific antibody-based assay” on line 71-72.

Reviewer 2 Report
This manuscript seems to contain some interesting results. However, it is difficult to understand and evaluate the paper due to the poor grammar. Please work with a professional editing company to improve the clarity and grammar of the manuscript. I have only pointed out a few of the many changes in wording that are needed.
line 38: Provide a reference for the 'Ohtsuka group'.
line 62: Change to '...have not been produced'.
line 74: Change to ‘…when placed near the functional domain…’.
Lines 94-95: ‘total nine pups were born from four mothers: four wild-type pups, six knock-in pups and three pups with deletions…’. 4 + 6 + 3 = 13 pups, not 9 pups.
line 102: What are ‘β mice’?
line 141: Change to ‘…shifted band compared to the wild-type band…’.
line 143: ‘mice’ in place of ‘mouse’.
lines 152-153: Change to ‘Asterisk shows shifted band of Flag-CaMKIIα protein compared to that of endogenous CaMKIIα protein’. Note the correct spelling of ‘asterisk’.
line 157: It is incorrect to say that antibodies were ’performed’. I suggest stating that ‘antibodies were used’.
line 165: Delete ‘were’.
Figure 5: Is ‘SNV chenge” correct or is that a typo?
line 192: Change to ‘…were shown in red color in the 3D structure…’.
line 202: Change ‘editing’ to ‘edited’.
line 207: Change to ‘…but it can only be used for detection of N…’.
line 212: Change to ‘i-GONAD is a powerful method with…’.
line 216: Change to ‘…is low at 15%...’.
lines 218-225: This section, as well as other parts of the manuscript, require a great deal of editing to improve the grammar and clarity of the presentation. Please work with a professional editing company to improve the quality of your writing.
line 229 – Change to ‘…from the 3D structure of the proteins’.
line 229: Change to ‘…both protein structures was…’.
line 245: Change ‘form’ to ‘from’.
line 257: What is meant by ‘was injected from the ovary side of oviduct’?
line 275: ‘reserved’? Do you mean ‘stored’?
lines 289-290: ‘Membrane were blocking’? What does this mean?
lines 308 and 315: I assume you mean ‘glass slides’.
line 336: ‘Data Availability Statement: Not applicable’. Why not?
Author Response
Reviewer #2 (Remarks to the Author):
We would like to thank the reviewer #2 for your instructive comments. We revised our manuscript according to your comments. We made corrections with "Track changes while editing" features of MS Word so that you can easily see which changes have been made.
This manuscript seems to contain some interesting results. However, it is difficult to understand and evaluate the paper due to the poor grammar. Please work with a professional editing company to improve the clarity and grammar of the manuscript. I have only pointed out a few of the many changes in wording that are needed.
We greatly appreciated your kind guidance. Our manuscript has been extensively edited by a native English editor at Editage (www.editage.com).
line 38: Provide a reference for the 'Ohtsuka group'.
We moved reference position form line 42 to line 40.
line 62: Change to '...have not been produced'.
We corrected it on line 66.
line 74: Change to ‘…when placed near the functional domain…’
We corrected it on line 78.
Lines 94-95: ‘total nine pups were born from four mothers: four wild-type pups, six knock-in pups and three pups with deletions…’. 4 + 6 + 3 = 13 pups, not 9 pups.
We corrected it on line 101.
line 102: What are ‘β mice’?
We corrected it to “Flag-CaMKIIβ on line 110”.
line 141: Change to ‘…shifted band compared to the wild-type band…’.
We corrected it on line 150.
line 143: ‘mice’ in place of ‘mouse’.
We corrected it on line 153.
lines 152-153: Change to ‘Asterisk shows shifted band of Flag-CaMKIIα protein compared to that of endogenous CaMKIIα protein’. Note the correct spelling of ‘asterisk’.
We corrected it on line 162-163.
line 157: It is incorrect to say that antibodies were ’performed’. I suggest stating that ‘antibodies were used’.
We corrected it to “immunostaining using anti-DYKDDDDK and anti-neuronal nuclei (NeuN) antibodies was performed“ on line 166-167.
line 165: Delete ‘were’.
We deleted it on line 177.
Figure 5: Is ‘SNV chenge” correct or is that a typo?
We corrected it to “SNV change” in Figure 5.
line 192: Change to ‘…were shown in red color in the 3D structure…’.
We corrected it on line 205.
line 202: Change ‘editing’ to ‘edited’.
We changed it on line 216.
line 207: Change to ‘…but it can only be used for detection of N…’.
We corrected it on line 221.
line 212: Change to ‘i-GONAD is a powerful method with…’.
We corrected it on line 227.
line 216: Change to ‘…is low at 15%...’.
We corrected on line 231.
lines 218-225: This section, as well as other parts of the manuscript, require a great deal of editing to improve the grammar and clarity of the presentation. Please work with a professional editing company to improve the quality of your writing.
To improve the quality of this section, we worked with Editage for English language editing.
line 229 – Change to ‘…from the 3D structure of the proteins’.
We corrected it on line 245.
line 229: Change to ‘…both protein structures was…’.
We corrected it on line 246.
line 245: Change ‘form’ to ‘from’.
We corrected it on line 265.
line 257: What is meant by ‘was injected from the ovary side of oviduct’?
We changed it to “injected into oviduct ampulla” on line 278-279.
line 275: ‘reserved’? Do you mean ‘stored’?
We changed it to “stored” on line 302.
lines 289-290: ‘Membrane were blocking’? What does this mean?
We changed it to “Membranes were blocked with” on line 318.
lines 308 and 315: I assume you mean ‘glass slides’.
We changed it to “glass slides” on line 337 and 344.
line 336: ‘Data Availability Statement: Not applicable’. Why not?
We changed it to “All data are available within the Article, Supplementary Information. The materials of this study are available from the corresponding author upon request.” on line 368-370.

Round 2
Reviewer 1 Report
The authors addressed most of my concerns.
Reviewer 2 Report
The clarity and quality of the presentation has been greatly improved due to the authors making the requested changes and also working with a professional editing company.